# Screening of qPCR Reference Genes in Quinoa Under Cold, Heat, and Drought Gradient Stress

**DOI:** 10.3390/plants14152434

**Published:** 2025-08-06

**Authors:** Qiuwei Lu, Xueying Wang, Suxuan Dong, Jinghan Fu, Yiqing Lin, Ying Zhang, Bo Zhao, Fuye Guo

**Affiliations:** Department of Biology, Xinzhou Normal University, Xinzhou 034000, China; luqw@xztu.edu.cn (Q.L.); 19956212964@163.com (X.W.); 15503412840@163.com (S.D.); 15969583046@163.com (J.F.); 18005025886@163.com (Y.L.); zying0709@163.com (Y.Z.); zhaobo050518@163.com (B.Z.)

**Keywords:** *Chenopodium quinoa*, RT-qPCR, reference genes, low-temperature stress, heat stress, drought stress, expression stability, gradient stress

## Abstract

Quinoa (*Chenopodium quinoa*), a stress-tolerant pseudocereal ideal for studying abiotic stress responses, was used to systematically identify optimal reference genes for qPCR normalization under gradient stresses: low temperatures (LT group: −2 °C to −10 °C), heat (HT group: 39° C to 45 °C), and drought (DR group: 7 to 13 days). Through multi-algorithm evaluation (GeNorm, NormFinder, BestKeeper, the ΔCt method, and RefFinder) of eleven candidates, condition-specific optimal genes were established as *ACT16* (Actin), *SAL92* (IT4 phosphatase-associated protein), *SSU32* (Ssu72-like family protein), and *TSB05* (Tryptophan synthase beta-subunit 2) for the LT group; *ACT16* and *NRP13* (Asparagine-rich protein) for the HT group; and *ACT16*, *SKP27* (S-phase kinase), and *NRP13* for the DR group, with *ACT16*, *NRP13*, *WLIM96* (LIM domain-containing protein), *SSU32*, *SKP27*, *SAL92*, and *UBC22* (ubiquitin-conjugating enzyme E2) demonstrating cross-stress stability (global group). *DHDPS96* (dihydrodipicolinate synthase) and *EF03* (translation elongation factor) showed minimal stability. Validation using stress-responsive markers—*COR72* (LT), *HSP44* (HT), *COR413-PM* (LT), and *DREB12* (DR)—confirmed reliability; *COR72* and *COR413-PM* exhibited oscillatory cold response patterns, *HSP44* peaked at 43 °C before declining, and *DREB12* showed progressive drought-induced upregulation. Crucially, normalization with unstable genes (*DHDPS96* and *EF03*) distorted expression profiles. This work provides validated reference standards for quinoa transcriptomics under abiotic stresses.

## 1. Introduction

Low temperatures, heat, and drought are major abiotic stresses that adversely affect crop growth and productivity by disrupting physiological, biochemical, and molecular processes, ultimately leading to reduced yield and quality [1,2]. Quinoa (*Chenopodium quinoa* Willd) exhibits notable tolerance to various abiotic stresses, conferring a degree of adaptability to extreme environments [3]. Despite this tolerance, low temperature, heat, and drought can still impose complex constraints on quinoa’s growth, physiology, and yield. Furthermore, quinoa is valued for its high protein content and balanced nutritional profile, including beneficial lipids, and for supplying sufficient amounts of all essential amino acids, including lysine, which is often limited in plant proteins [4]. Molecular investigations into quinoa have multiplied following the assembly and annotation of its reference genome [5,6,7]. The combination of quinoa’s nutritional excellence and relative resilience under challenging climatic and soil conditions highlights its significant potential for expanded production worldwide [4]. Understanding the molecular mechanisms underlying quinoa’s stress responses is crucial for harnessing this potential. However, stable reference genes must be identified and validated to conduct reliable gene expression studies under abiotic stress conditions. To date, no such validation of reference genes has been reported for gradient abiotic stress expression studies in *C. quinoa*.

The development of next-generation sequencing (NGS) technology has made it simpler and faster to analyze mRNA distribution and expression levels in the biosynthetic pathways of this plant [8]. In addition, another analysis method, quantitative real-time PCR (qPCR), is widely used in gene expression research due to its high sensitivity, quantitative accuracy, throughput capability, and low cost when using specific reference genes [9,10,11]. However, quantitative results can be influenced by many factors, such as genomic DNA contamination, RNA quality, primer specificity, and amplification efficiency [12]. To guarantee accurate results and minimize errors, using one or more stable and appropriate reference genes is essential. Ideally, reference gene expression should remain constant or vary minimally across tissues under different experimental treatments. However, it has been shown that the utility of reference genes must be validated for specific experimental conditions [13] and that normalization against a non-validated reference gene can compromise quantitative results [14]. Screening for adequate reference genes can be performed using statistical algorithms developed for this purpose. Scientists have developed several methods for systematic verification of reference genes, such as NormFinder [15], BestKeeper [16], geNorm [17], the ΔCt method [18], and RefFinder (http://blooge.cn/RefFinder/, accessed on 9 June 2025). [19,20], which integrate information on the expression of candidate reference genes and measure their relative stability. geNorm calculates an average expression stability value, defined as the average pairwise variation of a particular gene compared with all other potential reference genes. NormFinder identifies stably expressed genes based on a mathematical model that estimates both intra- and inter-group variation within the sample set. BestKeeper calculates the standard deviation (SD) and coefficient of variance (CV) as measures of stability. Reference genes suitable for different experimental conditions have been identified using these algorithms for a variety of crops, such as lettuce [21], pea [22], barley [23], okra [24], peach [13], *Dendrobium huoshanense* [25], maize [26], rice [27], *Ananas comosus* var. [28], garlic [29], and centipedegrass [30].

Based on transcriptome sequencing results and reported reference genes from other species, we selected eleven candidate reference genes belonging to two categories, including (1) seven genes with a low coefficient of variation (CV) screened from transcriptome data, namely IT4 phosphatase-associated protein (*SAL*), asparagine-rich protein (*NRP*), dihydrodipicolinate synthase 2 (*DHDPS*), LIM domain-containing protein (*WLIM*), S-phase kinase-associated protein (*SKP*), tryptophan synthase beta-subunit 2 (*TSB*), and Ssu72-like family protein (*SSU*), and (2) four traditional housekeeping genes involved in basic cellular processes and often reported to have stable expression, namely β-actin (*ACT*), ubiquitin-conjugating enzyme (*UBC*), and translation elongation factor (*EF*). The expression stability of these candidates in quinoa seedling leaves subjected to low temperature, heat, and drought treatments was assessed using the statistical algorithms geNorm, NormFinder, and BestKeeper and the ΔCt method. The online tool RefFinder was then employed to generate a comprehensive stability ranking. Finally, the selected reference genes’ reliability was validated through expression profiling analysis of target genes. This study aimed to identify suitable reference genes for molecular studies of abiotic stress responses in quinoa.

## 2. Results

### 2.1. Primer Specificity and Amplification Efficiency of Candidate Reference Genes

Based on the coefficient of variation derived from our laboratory’s existing RNA-seq data on quinoa leaves, we initially selected 11 candidate reference genes exhibiting stable expression (Table 1). Additionally, *COR72*, *HSP44*, *COR413-PM*, and *DREB12* were chosen as target genes for validating the reference genes based on transcriptome differential gene expression analysis.

Primer specificity was confirmed using agarose gel electrophoresis and melting curve analysis. The electrophoresis results demonstrated a single distinct band for the PCR product of each of the 15 genes. The melting curve analysis revealed a single specific peak for each primer pair (Figure 1). These results indicate that the primers for all 15 genes exhibited good specificity, meeting the requirements for subsequent gene expression analysis.

Amplification efficiency analysis was performed using templates prepared by pooling cDNA from quinoa leaves subjected to low-temperature (LT), heat (HT), and drought (DR) stress treatments, as well as from control (CK) samples. This pooled cDNA was serially diluted in 10-fold increments (10^0^ to 10^−4^). Analysis showed that the amplification efficiencies for the 15 primer pairs ranged from 94.93% (*DHDPS96*) to 108.03% (*SKP27*), with correlation coefficients (R^2^) all exceeding 0.98 (Table 1), fulfilling standard qPCR requirements. These data demonstrate that the selected reference gene primer system is stable and reliable and thus suitable for quantitative analysis.

### 2.2. Expression Levels of Candidate Reference Genes

The cycle threshold (Ct) value directly reflects the gene expression levels, with lower Ct values indicating higher expression. Four conditions were used, defined as follows:

The global group: control (CK), low-temperature (LT), heat (HT), and drought (DR) samples; the LT group: control (CK) and low-temperature (LT) samples; the HT group: control (CK) and heat (HT) samples; and the DR group: control (CK) and drought (DR) samples.

Specifically, *UBC22* consistently showed the highest expression (lowest mean Ct) in the global, low-temperature (LT), and drought (DR) groups, while *UBC19* exhibited the lowest expression (highest mean Ct) in these groups. Under heat (HT) stress, *ACT16* demonstrated the highest expression level, whereas *SSU32* displayed minimal expression.

The raw Ct values of the 15 genes are shown in Appendix A and plotted on a boxplot (Figure 2). Notably, some genes showed consistent expression across conditions, while others exhibited marked variability, underscoring the importance of validating reference gene stability within specific experimental contexts. A boxplot analysis revealed substantial variation in expression stability among candidate genes under LT, HT, and DR stresses, as evidenced by differential interquartile ranges. Comprehensive stability ranking identified *NRP13* as the most stable reference gene globally and under HT/DR conditions, while *UBC22* showed superior stability in the LT group. The complete stability hierarchies were as follows: global group (Figure 2a): *NRP13* > *ACT16* > *UBC22* > *WLIM96* > *SSU32* > *SKP27* > *TSB05* > *EF03* > *UBC19* > *SAL92* > *DHDPS96*; LT group (Figure 2b)**:** *UBC22* > *NRP13* > *ACT16* > *SAL92* > *WLIM96* > *SKP27* > *TSB05* > *SSU32* > *EF03* > *DHDPS96* > *UBC19*; HT group (Figure 2c): *NRP13* > *ACT16* > *WLIM96* > *UBC22* > *TSB05* > *SKP27* > *SSU32* > *UBC19* > *EF03* > *SAL92* > *DHDPS96*; DR group (Figure 2d): *NRP13* > *TSB05* > *SSU32* > *WLIM96* > *ACT16* > *SAL92* > *SKP27* > *UBC22* > *DHDPS96* > *UBC19* > *EF03*.

### 2.3. Analysis of the Expression Stability of Candidate Reference Genes

To reliably assess the expression stability of the 11 candidate reference genes, we utilized RefFinder, a comprehensive online tool that combines the four most popular algorithms for reference gene evaluation: geNorm, NormFinder, BestKeeper, and the comparative ∆Ct method. This methodology allowed us to produce individual rankings from each algorithm and an integrated overall ranking based on the geometric mean of the ranks obtained from each method (Figure 3, Appendix A).

Stability rankings were independently generated for each treatment condition (LT, HT, and DR) alongside a unified global ranking across all conditions (Figure 3a). The findings indicate variability in the stability of the assessed genes based on the treatment, ranging from most stable to least stable. For the LT group (Figure 3b), the optimal genes of the candidate reference genes were *ACT16* and *SAL92*, while for the HT group (Figure 3c), the best genes were *ACT16* and *NRP13*, and the least stable gene for the LT and HT groups was *DHDPS96*. Moreover, the optimal genes for DR treatment were *ACT16* and *SKP27*, while the least stable gene was *EF03* (Figure 3d). It is important to note that some genes displayed consistent expression across treatments. For instance, *ACT16* ranked among the most stable genes in the LT, HT, and DR groups.

A comprehensive analysis integrating data from all methods was also included. The global analysis proposed the following ranking from most to least stable: *ACT16*, *NRP13*, *SSU32*, *WLIM96*, *UBC22*, *SAL92*, *SKP27*, *TSB05*, *EF03*, *UBC19*, and *DHDPS96*. Based on these findings, *ACT16* was the most suitable internal control for normalizing gene expression data across all evaluated abiotic stresses.

### 2.4. Analysis of Candidate Reference Gene Expression Stability Using GeNorm

GeNorm evaluates the expression stability of genes by calculating their stability measure (M-value). Lower M-values indicate greater stability. The average M-value progressively decreased as less stable candidate genes were sequentially excluded from the analysis (Figure 4a). The pairwise combination of genes with the lowest M-value (i.e., the most stable reference genes) was identified as follows: global group: *ACT16* and *NRP13*; LT group: *ACT16* and *SAL92*; HT group: *ACT16* and *NRP13*; and DR group: *ACT16* and *SKP27* (Appendix A).

To determine the optimal number of genes needed for reliable normalization, we used the geNorm tool. It employs pairwise variation analysis to calculate the V_n/n+1_ value, which compares the stability of using n genes versus n + 1 genes. A cutoff value of 0.15 is suggested: if V_n/n+1_ < 0.15, no additional genes are necessary; however, if V_n/n+1_ ≥ 0.15, more genes should be included for normalization. The results show that the V_2/3_ value for HT group was 0.113, the V_3/4_ value for the DR group was 0.135; the V_4/5_ value for the LT group was 0.136; and the V_7/8_ value for the global group was 0.135 (Figure 4b). This suggests that two reference genes were sufficient for robust normalization under the HT group and three reference genes were sufficient for robust normalization under the DR group. Four reference genes were sufficient for the LT group, and seven reference genes were sufficient for the global group. Therefore, it is recommended to use the corresponding number of genes under different stresses to improve the reliability of gene expression analysis.

### 2.5. Validation of Candidate Reference Genes

To validate the condition-specific optimal reference genes, we analyzed the expression patterns of target genes *COR72*, *HSP44*, *COR413-PM*, and *DREB12* using distinct normalization strategies. For low-temperature (LT) stress validation, single-reference normalizations employed the most stable genes (*ACT16* and *SAL92*) and least stable gene (*DHDPS96*), while multi-gene normalizations used the LT combination (*ACT16*, *SAL92*, *SSU32*, and *TSB05*) and global combination (*ACT16*, *NRP13*, *WLIM96*, *SSU32*, *SKP27*, *SAL92*, and *UBC22*). Both cold-responsive genes exhibited significant induction under gradient LT stress but with distinct oscillatory patterns: *COR72* showed a biphasic “increase–decrease–increase–decrease” response with peaks at −4 °C and −8 °C (Figure 5a, Appendix A), whereas *COR413-PM* displayed an “increase–decrease–increase–decrease–increase” trend, peaking at −2 °C and −6 °C (Figure 5b, Appendix A). Notably, the *COR72* expression profiles normalized to the LT and global combinations demonstrated higher congruence than single-gene normalizations.

For heat (HT) validation, normalizations included the optimal pair (*ACT16* and *NRP13*), least stable gene (*DHDPS96*), HT combination (*ACT16* and *NRP13*), and global combination. *HSP44* expression progressively increased from 39 °C to 43 °C, followed by attenuation at 45 °C (Figure 5c, Appendix A). Crucially, normalization with *ACT16*, *NRP13*, or their combination yielded concordant expression trajectories, confirming the reliability of HT-specific reference genes.

Drought stress validation employed the optimal pair (*ACT16* and *SKP27*), least stable gene (*EF03*), DR combination (*ACT16*, *SKP27*, and *NRP13*), and global combination. *DREB12* exhibited progressive upregulation with prolonged drought. Consistent expression patterns emerged across normalizations using *ACT16*, *SKP27*, the DR combination, and the global combination (Figure 5d, Appendix A). Strikingly, normalization with the least stable genes (*DHDPS96* for the LT/HT groups; *EF03* for the DR group) substantially distorted target gene expression profiles across all stress regimes.

## 3. Discussion

Abiotic stresses, including cold, heat, salinity, and drought, constrain plant growth and yield. Plants have evolved multifaceted mechanisms to mitigate stress-induced damage, involving intricate transcriptional reprogramming and post-translational regulation [31]. In gene expression studies, normalization using stably expressed reference genes (internal controls) is critically important—particularly for stress-tolerant species. Chenopodiaceae plants are renowned for their tolerance to drought, salinity, and cold. It has been reported that systematic screening of reference genes in the Chenopodiaceae annual herb Salsola ferganica under abiotic stress has laid a foundation for gene expression studies under such conditions [32]. Likewise, gene expression research on quinoa under cold, heat, and drought stress requires identifying reference genes with maximum stability under these specific conditions to ensure experimental accuracy [33]. Given that reference gene stability varies across tissue types and experimental treatments, condition-specific validation remains imperative.

While qPCR offers high sensitivity and specificity for gene expression analysis, its reliability crucially depends on appropriate reference genes for data normalization. Current studies on quinoa reference genes remain limited, with reports only covering salt stress [34], diurnal rhythms [35], and downy mildew infection [36]. Despite quinoa’s renowned stress tolerance [37], varietal differences in stress responses exist. For instance, metabolomic dynamics under cold stress differ significantly between cultivars Dian Quinoa 2324 and Dian Quinoa 281 [38]. This genetic diversity necessitates selecting reference genes demonstrating cross-cultivar stability. Conventional single-intensity stress treatments may yield false positive results due to uncontrolled compensatory mechanisms. To circumvent this limitation, we implemented gradient stress exposures spanning physiologically relevant thresholds. This approach more accurately captures the dynamic stability profiles of reference genes and reveals authentic biological functions of target genes. To date, no systematic investigation has identified reference genes for quinoa subjected to gradient cold, heat, or drought treatments—a knowledge gap addressed by this study.

Quantitative real-time PCR (qRT-PCR) yielded Ct values for candidate reference genes. Optimal reference genes should exhibit expression levels comparable to target genes, with Ct values typically ranging from 15 to 30 cycles. Our analysis confirmed that all 11 candidate reference genes met this criterion (Ct = 15–30), enabling subsequent stability assessment. Divergent stability rankings emerged across analytical methods (geNorm, NormFinder, the ΔCt method, and BestKeeper), attributable to distinct statistical methodologies. Such discrepancies are well-documented: Zhuang et al. [39] reported algorithm-dependent optimal gene selection in *Oxytropis ochrocephala* Bunge studies, while Fan et al. [40] observed inconsistent rankings across three programs when analyzing bamboo tissues. These observations underscore the limitation of single-algorithm approaches, which may yield biased stability assessments [41]. Consequently, we employed a multi-algorithm framework integrating geNorm, NormFinder, the ΔCt method, and BestKeeper.

Divergent stability rankings emerged across analytical software (geNorm, NormFinder, BestKeeper, and the ΔCt method) due to fundamentally distinct computational algorithms. This methodological variability is well-established: while BestKeeper identified *UBC22* and *WLIM96* as the most stable genes across all samples, geNorm and NormFinder consistently prioritized *ACT16* and *NRP13* for the same condition. Such algorithm-dependent discrepancies highlight the absence of consensus regarding optimal stability assessment methods. To resolve this issue, we employed the online tool RefFinder. It integrates the currently available major computational programs (geNorm, Normfinder, BestKeeper, and the comparative ΔCt method) to compare and rank the tested candidate reference genes. Based on the rankings from each program, it assigns an appropriate weight to an individual gene and calculated the geometric mean of their weights for the overall final ranking [19].

Cold-induced proteins (COR) exhibit crucial expression dynamics under low-temperature stress in plants, particularly well-characterized in the model species *Arabidopsis thaliana*. Heterologous expression studies demonstrate that transferring *COR* genes enhances cold tolerance: *COR413PM2* from *Phlox subulata* improved freezing resistance in transgenic *Arabidopsis* [42], while overexpression of *CpCOR413-PM1* from wintersweet (*Chimonanthus praecox*) increased cold hardiness [43]. Consistent with these findings, both *COR72* and *COR413PM* in quinoa displayed pronounced cold-responsive expression patterns in our study. And the response of *COR413PM* to low temperatures is weaker than *COR72* (Figure 5a,b, Appendix A). Compared to the control samples (CK), *COR72* exhibited upregulation, initiating at −2 °C and peaking at −8 °C, and maintained stable expression from −4 °C to −10 °C. In contrast, *COR413-PM* showed selective induction with significant upregulation only at −2 °C and −6 °C while demonstrating downregulation at other temperature points. This divergence suggests distinct regulatory mechanisms for these cold-responsive genes. For heat stress validation, we selected heat shock proteins (HSPs)—evolutionarily conserved molecular chaperones that maintain proteostasis under thermal stress. HSPs prevent protein aggregation and assist in the refolding/degradation of misfolded proteins [44]. Under gradient heat stress, quinoa *HSPs* exhibited significant upregulation, peaking at 43 °C, beyond which expression declined (Figure 5c, Appendix A). This attenuation likely reflects thermal denaturation or degradation of HSPs themselves under extreme temperatures. Drought-responsive mechanisms involve dehydration-responsive element-binding (DREB) transcription factors that orchestrate downstream gene networks to enhance osmotic stress tolerance [45]. Our data confirmed progressive *DREB12* upregulation with intensifying drought severity, consistent with its role in activating physiological adaptations [46]. Crucially, expression profiling of stress marker genes—*COR72* (LT), *HSP44* (HT), *COR413PM* (LT), and *DREB12* (DR)—using condition-specific reference gene combinations validated the reliability of our normalization strategy across all stress regimes.

Crucially, normalization with multiple reference genes significantly reduces experimental error inherent to single-gene standardization [47,48]. Our pairwise variation (V_n/n+1_) analysis determined that four reference genes were required for reliable normalization under LT stress. This multi-gene approach stabilized *COR72* expression across −4 °C to −10 °C, eliminating statistically significant differences within this temperature range (*p* > 0.05) that persisted when normalized to single genes or global combinations. Notably, *COR413-PM* consistently exhibited significant upregulation at −6 °C regardless of the normalization strategy, though LT combination normalization amplified its fold change magnitude, demonstrating superior sensitivity. Under HT stress, global combination normalization distorted *HSP44* expression trajectories compared to single-gene or HT-specific normalization, with significant divergence at 41 °C (*p* < 0.05), confirming the appropriateness of condition-specific references. For drought stress, all normalization methods yielded comparable results, except for minor significance variations with *ACT16* alone at 7d (Appendix A).

While V_n/n+1_ analysis provides a robust framework for determining optimal gene numbers, discrepancies emerged between GeNorm and RefFinder stability rankings. Under the global treatment (Figure 4), V_6/7_ and V_7/8_ both fell below the 0.15 threshold (0.142 and 0.135, respectively), yet GeNorm and RefFinder reversed the rankings of the sixth and seventh genes. Consequently, we included seven reference genes in the global combination to reconcile algorithmic conflicts. This harmonization effect—evident in the expression patterns of *COR72*, *HSP44*, *COR413-PM*, and *DREB12*—demonstrates how multi-gene normalization yields weighted intermediate profiles that mitigate individual reference gene biases and enhance analytical accuracy.

## 4. Materials and Methods

### 4.1. Plant Materials

JL3 (*Chenopodium quinoa*) plants were cultivated in a growth chamber at 24 °C under a 12 h light/12 h dark photoperiod. At six weeks of age, the plants were subjected to low-temperature, heat, and drought treatments (experimental groups and specific treatment methods are detailed in Table 2). Each treatment group (including each stress gradient and the control) consisted of three biological replicates (i.e., three individual pots of plants). After flash freezing in liquid nitrogen, the samples were stored at −80 °C.

### 4.2. RNA Extraction and cDNA Synthesis

The total RNA was extracted from the samples. RNA concentration and purity were measured using a QuickDrop^TM^ Micro-volume UV-Vis Spectrophotometer (Molecular Devices, San Jose, CA, USA). RNA samples meeting the criteria of OD_260_/OD_280_ ratios between 1.8 and 2.2 and OD_260_/OD_230_ ratios > 1.8 were deemed suitable for subsequent procedures. First-strand cDNA was synthesized using the Might-yScript^TM^ Plus First Strand cDNA Synthesis Kit (Sangon Biotech, Shanghai, China) and stored at −20 °C.

### 4.3. Candidate Gene Selection and Primer Design

Eleven candidate reference genes were selected. Additionally, four target genes—cold-induced protein (*COR72*), heat shock protein (*HSP44*), cold-induced protein (*COR413-PM*), and dehydration-responsive element-binding protein (*DREB12*)—were chosen to validate reference gene stability (Table 3). Primers were designed using Primer Premier 5.0 software. All primers were synthesized by Sangon Biotech Co., Ltd. (Shanghai, China)

### 4.4. Primer Specificity Verification and qPCR Analysis

cDNA synthesized from normally cultivated quinoa leaves served as the template to verify the specificity of the primers listed in Table 3 via conventional PCR amplification. The PCR reaction mixture (20 μL total volume) contained 1 μL each of forward and reverse primers (10 μmol/L), 2 μL of cDNA (1 μg), 6 μL of ddH_2_O, and 10 μL of 2 × Taq DNA Polymerase Premix. The PCR cycling conditions were as follows: initial denaturation at 95 °C for 5 min, followed by 40 cycles of denaturation at 95 °C for 15 s and annealing/extension at 60 °C for 30 s, with a final hold at 4 °C. PCR products were analyzed using 1% agarose gel electrophoresis to confirm the amplicon size and assess the presence of non-specific amplification bands.

cDNA synthesized from quinoa leaves subjected to low-temperature, heat, and drought treatments served as the template for quantitative real-time PCR (qPCR) analysis using a CFX96 Real-Time PCR Detection System (Bio-Rad, Hercules, CA, USA). The qPCR reaction mixture (20 μL total volume) contained 10 μL of 2 × SYBR Green Premix, 0.8 μL each of forward and reverse primers (10 μmol/L), 2 μL of cDNA (1 μg), and 6.4 μL of ddH_2_O. The qPCR cycling conditions were as follows: initial denaturation at 95 °C for 3 min, followed by 40 cycles of denaturation at 95 °C for 5 s and annealing/extension at 60 °C for 30 s. A melt curve analysis was subsequently performed by heating the amplification products from 65 °C to 95 °C.

### 4.5. Primer Amplification Efficiency Analysis

The amplification efficiency of the primers was analyzed using qPCR with a serially diluted mixed cDNA template. Specifically, cDNA synthesized from quinoa leaves subjected to low-temperature, heat, and drought treatments was pooled and serially diluted from 10^0^ (undiluted) to 10^−4^. Using this mixture as the template, qPCR amplification was performed for each of the 11 candidate reference genes following the protocol described in Section 4.4.

The linear relationship between the log-transformed cDNA template concentration and the Ct value was analyzed using Microsoft Excel software (Version 2408, Build 16.0.17928.20100, Access on June 9, 2025). The linear correlation coefficient (R^2^) and slope (S) of the standard curve were determined. The amplification efficiency (E) was calculated using the following formula:E (%) = (10^(−1/S)^ − 1) × 100%(1)

### 4.6. RT-qPCR Data Analysis of Reference Gene Stability

Gene expression stability was evaluated as follows: First, the mean Ct value was calculated from three biological replicates to analyze the expression levels of the 11 candidate reference genes. Subsequently, the expression stability of these genes was assessed using four statistical algorithms: GeNorm, NormFinder, BestKeeper, and the ΔCt method. Finally, the results from these multiple algorithms were integrated using the online tool RefFinder (http://blooge.cn/RefFinder/, accessed on 9 June 2025) to generate a comprehensive stability ranking.

The optimal reference genes were screened in two distinct groups across all samples. For the global group, screening was performed based on all treatments detailed in Table 1 (CK, LT, HT, and DR). Within specific stress treatments, screening was performed for optimal reference genes specifically under LT, HT, and DR conditions.

### 4.7. Normalization of Stress-Response Gene Expression in Chenopodium quinoa by RT-qPCR Analysis

To validate the selected reference genes, the relative expression levels of the target genes *COR72*, *HSP44*, *COR413-PM*, and *DREB12* were calculated using the 2^−ΔΔCt^ method. This analysis was performed using the following distinct reference gene combinations: the two most stable reference genes overall; the least stable reference gene; the most stable reference gene combination identified across all samples (global); and the most stable reference gene combination identified specifically for each individual stress treatment (LT, HT, or DR).

### 4.8. Data Processing and Accessibility

All experimental results are three independent replicates (Three biological replicates and three technical replicates). The data were analyzed using the IBM SPSS Statistics software (version 21.0), statistical significance was assessed using one-way ANOVA followed by LSD test. Different letters denote significance levels: *p* < 0.05. RNAseq reads were deposited at NCBI under the Experiment codes: SRR5572144, SRR5572145, SRR5572148, SRR5572157, SRR5572158, SRR5572160, SRR5572163-SRR5572174, SRR34789946-SRR34789951, SRR34789956, SRR34789967, and SRR34789968.

## 5. Conclusions

This study systematically evaluated 11 candidate reference genes in quinoa under abiotic stresses using a multi-algorithm framework (GeNorm, NormFinder, BestKeeper, the ΔCt method, and RefFinder). Condition-specific optimal reference gene pairs were determined as follows: *ACT16*, *SAL92*, *SSU32*, and *TSB05* for low-temperature (LT) stress; *ACT16* and *NRP13* for heat (HT) stress; and *ACT16*, *SKP27*, and *NRP13* for drought (DR) stress. *ACT16*, *NRP13*, *WLIM96*, *SSU32*, *SKP27*, *SAL92*, and *UBC22* demonstrated cross-stress stability across all conditions. Validation through expression profiling of stress-responsive target genes (*COR72*, *HSP44*, *COR413-PM*, and *DREB12*) confirmed the reliability of these reference genes, whereas normalization with the least stable genes (*DHDPS96* and *EF03*) abolished stress-responsive patterns. This study establishes validated reference standards for quinoa gene expression studies, providing critical molecular tools for deciphering abiotic stress adaptation mechanisms.

## Figures and Tables

**Figure 1 plants-14-02434-f001:**
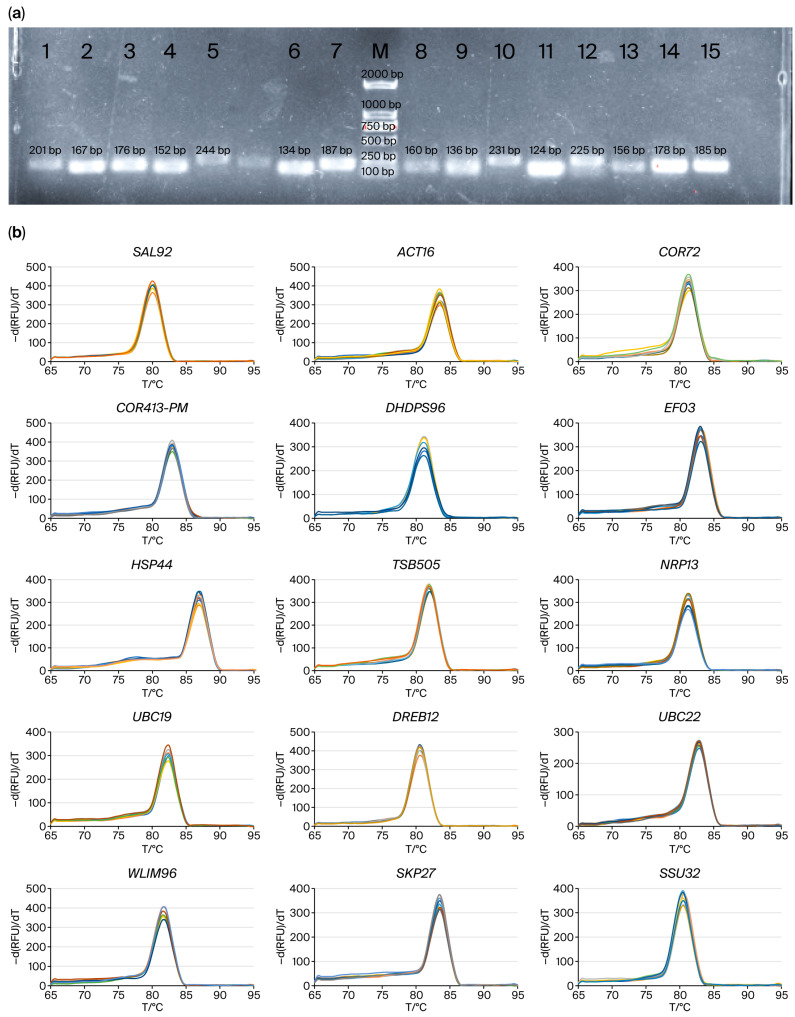
Specificity analysis of primers for candidate reference and target genes. (**a**) Agarose gel electrophoresis of PCR products of 11 candidate reference genes and 4 target genes in quinoa. M: DL2000 DNA Marker; 1: *DHDPS96*: dihydrodipicolinate synthase 2 (201 bp); 2: *SAL92*: IT4 phosphatase-associated protein (167 bp); 3: *ACT16*: actin (176 bp); 4: *NRP13*: asparagine-rich protein (152 bp); 5: *UBC19*: ubiquitin-conjugating enzyme E2 (244 bp); 6: *EF03*: translation elongation factor (134 bp); 7: *WLIM96*: LIM domain-containing protein (187 bp); 8: *SKP27*: S-phase kinase-associated protein (160 bp); 9: *COR72*: cold-regulated protein (136 bp); 10: *HSP44*: heat shock protein (231 bp); 11: *COR413-PM*: cold-regulated 413-plasma membrane protein (124 bp); 12: *DREB12*: dehydration response element binding protein (225 bp); 13: *TSB05*: tryptophan synthase beta-subunit 2 (156 bp); 14: *SSU32*: Ssu72-like family protein (178 bp); 15: *UBC22*: ubiquitin-conjugating enzyme E2 (185 bp). (**b**) Melting curves of 11 candidate reference genes and 4 target genes in quinoa.

**Figure 2 plants-14-02434-f002:**
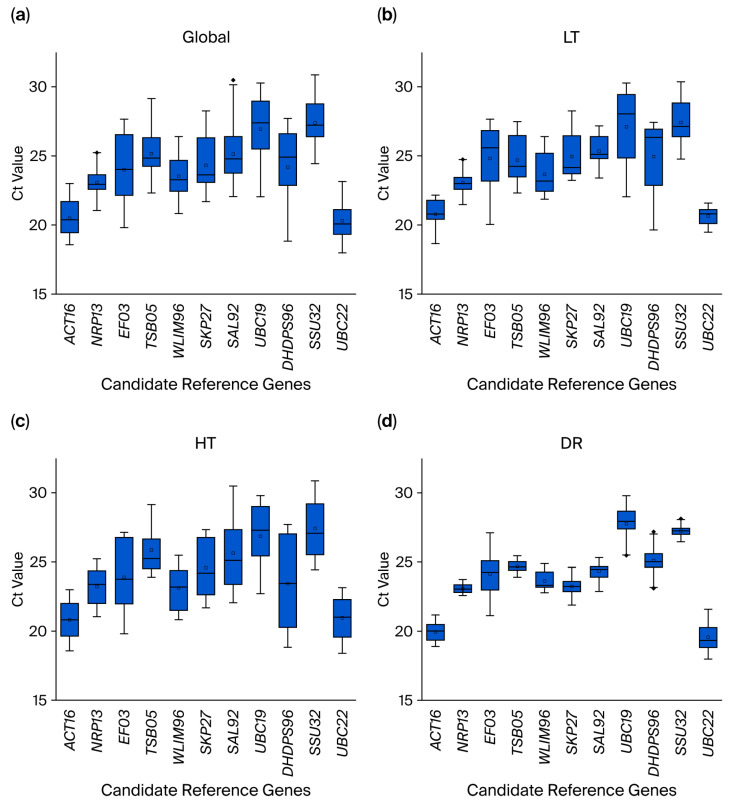
A comparison of cycle threshold (Ct) values for the 11 candidate reference genes in samples subjected to different treatments. (**a**) Global group: Control (CK), low-temperature (LT), heat (HT), and drought (DR). (**b**) LT group: Low-temperature treatment group. (**c**) HT group: Heat treatment group. (**d**) DR group: Drought treatment group.

**Figure 3 plants-14-02434-f003:**
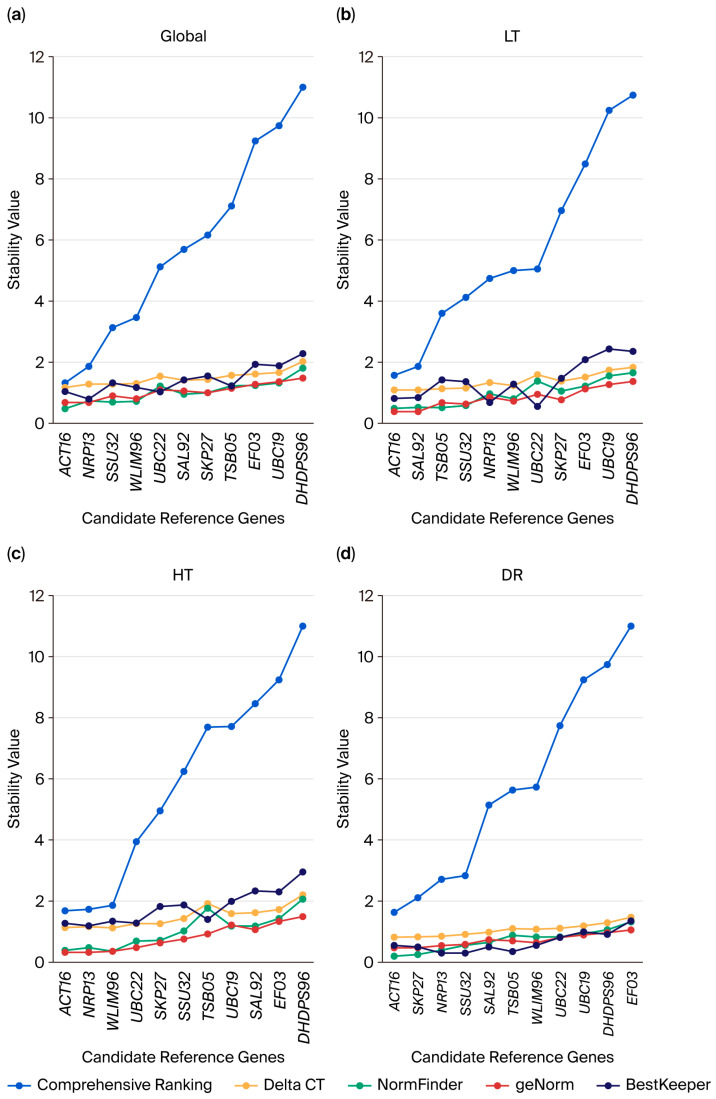
Analysis of expression stability of candidate reference genes using multiple software. (**a**) Global group: Control (CK), low-temperature (LT), heat (HT), and drought (DR). (**b**) LT group: Low-temperature treatment group. (**c**) HT group: Heat treatment group. (**d**) DR group: Drought treatment group.

**Figure 4 plants-14-02434-f004:**
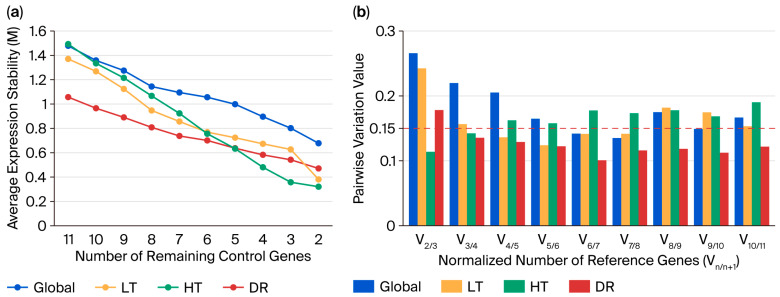
Candidate reference genes assayed using GeNorm. (**a**) Expression stabilities of candidate reference genes. (**b**) Normalized number of reference genes. Global group: Control (CK), low-temperature (LT), heat (HT), and drought (DR). LT group: Low-temperature treatment group. HT group: Heat treatment group. DR group: Drought treatment group.

**Figure 5 plants-14-02434-f005:**
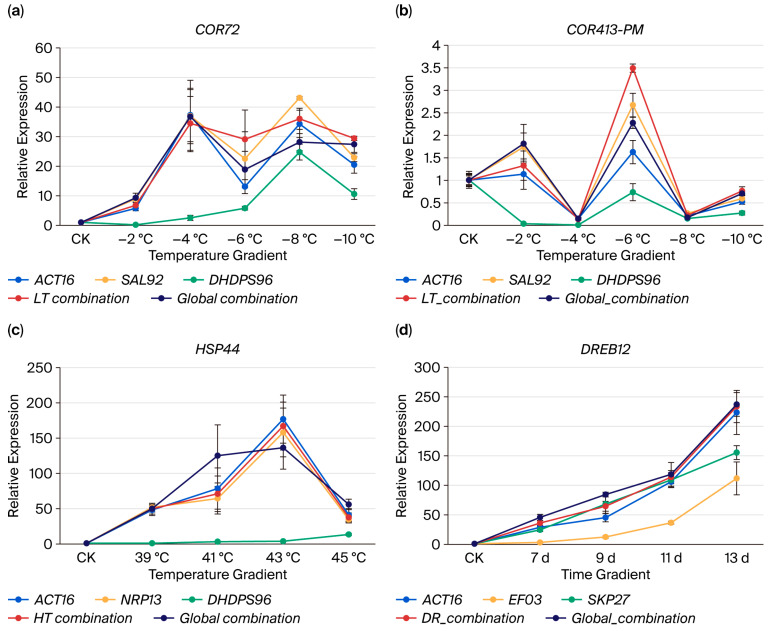
Expression pattern analysis of target genes with different candidate genes as reference genes. (**a**) *COR72*. (**b**) *COR413-PM*. (**c**) *HSP44*. (**d**) *DREB12*. LT combination: *ACT16*, *SAL92*, *SSU32*, and *TSB05*. HT combination: *ACT16* and *NRP13*. DR combination: *ACT16*, *SKP27*, and *NRP13*. Global combination: *ACT16*, *NRP13*, *WLIM96*, *SSU32*, *SKP27*, *SAL92*, and *UBC22*.

**Table 1 plants-14-02434-t001:** Amplification efficiency of candidate reference genes and target genes.

Gene	Coefficient of Variation (CV)	Amplification Efficiency/%	Linear Correlation Coefficient (R^2^)
*SAL92*	0.0208 (LT)	99.02%	0.9881
*ACT16*	0.0411 (LT)	96.66%	0.9891
*DHDPS96*	0.0412 (HT)	94.93%	0.9877
*EF03*	0.0876 (HT)	104.03%	0.9961
*TSB05*	0.0417 (HT)	102.34%	0.9986
*NPR13*	0.0784 (DR)	104.36%	0.9911
*UBC19*	0.0463 (DR)	102.83%	0.9978
*UBC22*	0.0853 (Global)	96.97%	0.9967
*WLIM96*	0.0806 (Global)	98.04%	0.9837
*SKP27*	0.0829 (Global)	108.03%	0.9879
*SSU32*	0.0891 (Global)	97.33%	0.9901
*COR72*	1.2999 (Global)	98.72%	0.9835
*COR413-PM*	0.5632 (Global)	98.49%	0.9919
*HSP44*	0.9073 (Global)	102.31%	0.9880
*DREB12*	0.6486 (Global)	99.30%	0.9900

**Table 2 plants-14-02434-t002:** Experimental grouping and treatments.

Group	Experiment Treatments (Leaves Were Collected from Six-Week-Old Quinoa)
Normal treatment (CK)	Plants were continuously maintained under normal growth conditions (24 °C, 12 h light/12 h dark cycle, with regular watering) in the growth chamber.
Low-temperature treatment (LT)	Five temperature gradients were set: −2 °C, −4 °C, −6 °C, −8 °C, and −10 °C. Upon reaching the target temperature, plants were transferred from the growth conditions (24 °C) to a low-temperature incubator. Plants were subjected to stress treatment for 6 h at a constant temperature.
Heat treatment (HT)	Four temperature gradients were set: 39 °C, 41 °C, 43 °C, and 45 °C. Plants were transferred directly from growth conditions (24 °C) into a growth chamber that had been pre-set and stabilized at the target temperature. Plants were subjected to stress treatment for 6 h at a constant temperature.
Drought treatment (DR)	Four stress gradients were set based on the duration after watering cessation: 7, 9, 11, and 13 days. The day when watering was stopped was denoted as day 0 of drought treatment. Drought treatment was conducted within the growth chamber (24 °C, 12 h light/12 h dark).

**Table 3 plants-14-02434-t003:** Candidate reference gene and primer information.

Gene ID	Gene	Description	Function	Primer Sequence (5′~3′)	Product Size
AUR62038592	*SAL92*	IT4 phosphatase-associated protein	It is required for SIT4’s role in G1 cyclin transcription and for bud formation.	F: GAACACTCACATAGCACCTT R: CGAACCAACACCTCCATA	167 bp
AUR62019116	*ACT16*	Actin	Actin is a ubiquitous protein involved in the formation of filaments that are major components of the cytoskeleton.	F: TTGTGCTCAGTGGTGGTA R: CATCTGTTGGAAGGTGCT	176 bp
AUR62003513	*NRP13*	Asparagine-rich protein	It plays a role in phytohormone response, embryo development and programmed cell death by pathogens or ozone.	F: GAACAAGCCGGAATGTAA R: AAATAAACCCAAGCCAGA	152 bp
AUR62036119	*UBC19*	Ubiquitin-conjugating enzyme E2	It acts as a ubiquitin-binding enzyme	F: ATTGATAAGCTAGGGAGG R: AGAGGGTAAAGTTGTTGC	244 bp
AUR62021096	*DHDPS96*	Dihydrodipicolinate synthase 2	Key enzymes of lysine biosynthesis pathway	F: CTTTACAAACGCCACCAT R: GAGAAGCAGAGCGAGGAC	201 bp
AUR62026903	*EF03*	Translation elongation factor	Catalyzes the GTP-dependent ribosomal translocation step during translation elongation	F: CCGCACTGTGATGAGCAA R: TGGAACGAACCTTGGGAT	134 bp
AUR62012196	*WLIM96*	LIM domain-containing protein	The exact function is unknown	F: ACAAGGTCGCCAAGCAAA R: TTCCATCAAGGGCAGCAT	187 bp
AUR62013027	*SKP27*	S-phase kinase-associated protein	Participate in the ubiquitination of target proteins and subsequent proteasomal degradation	F: TTTTGGCTGCTAACTACCT R: TTCTCCCTCCTAACCTCC	160 bp
AUR62012972	*COR72*	Cold-regulated protein	Participate in low temperature response	F: GGTAGACAAGGCAGAGGA R: TGTAGGCTGATGATGGTTAT	136 bp
AUR62021644	*HSP44*	Heat shock protein	Molecular chaperones that suppress protein aggregation and protect against cell stress	F: CCTCGCACAGTCCCATAC R: CAACTCAGCCTTCGCATC	231 bp
AUR62016670	*COR413-PM*	Cold-regulated 413-plasma membrane protein	Participate in low temperature response	F: AGCATCCTATGTCCGTGGTG R: CCCGTTAGCCCTTGTGAA	124 bp
AUR62012312	*DREB12*	Dehydration response element binding protein	Participate in plant drought stress	F: ACTTGCCGCATTACCCAG R: GCATCATCGCAGCATTTT	225 bp
AUR62020505	*TSB05*	Tryptophan synthase beta-subunit 2	Catalyzes the final step in the biosynthesis of L-tryptophan	F: TCTGAAAGACTTGGGACG R: TTCGGAAGAGTTGGACAC	156 bp
AUR62036432	*SSU32*	Ssu72-like family protein	It has intrinsic phosphatase activity and plays an essential role in the transcription cycle	F: CCTCAACGCTGGCAAGAT R: CACCAATAGCCGCCTCCT	178 bp
AUR62028822	*UBC22*	Ubiquitin-conjugating enzyme E2	It acts as a ubiquitin-binding enzyme	F: AAGAGGTTGATGAGGGAT R: GGAGGCTTATTTGGGTAG	185 bp

## Data Availability

The sequence of the genes used in the RT-qPCR experiments can be made available upon request.

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
