# Peer review of "Screening of qPCR Reference Genes in Quinoa Under Cold, Heat, and Drought Gradient Stress"

_plants, 2025, doi:10.3390/plants14152434_

Round 1
Reviewer 1 Report
Comments and Suggestions for Authors
The manuscript by Lu et al is a well-structured and scientifically relevant study focused on the selection and validation of reference genes for Chenopodium quinoa under three types of abiotic stress: low temperature, heat, and drought. The work is timely and valuable, particularly considering the increasing importance of quinoa as a resilient crop under climate change conditions.
One of the main strengths of the study lies in its methodological rigor. The authors utilize a comprehensive pipeline to evaluate reference gene stability, including four widely accepted algorithms—geNorm, NormFinder, BestKeeper, and the ΔCt method—followed by integration using the RefFinder tool. This multi-algorithm approach enhances the robustness of the results and minimizes the bias often associated with single-method evaluations. Additionally, the validation step using stress-responsive marker genes (COR72, HSP44, COR413-PM, DREB12) adds reliability to the conclusions and demonstrates the practical impact of gene selection on downstream gene expression studies.
The study will be useful for researchers conducting gene expression analysis in quinoa and potentially in other similar crop but some small concerns have to be addressed before publication.
- A thorough revision at the linguistic and editorial levels is needed.
- The statistical methods used to determine significance (p-values) are mentioned in passing but not described. It would be helpful to specify whether ANOVA, t-tests, or other methods were used. Additionally, although the introduction mentions the existence of cultivar-specific stress responses, the manuscript does not clearly state which quinoa variety was used for the experiments. This information is important, especially for assessing reproducibility and applicability.
To me opinion, English needs to be revised
Author Response
Reviewer 1 Comments:
The study will be useful for researchers conducting gene expression analysis in quinoa and potentially in other similar crop but some small concerns have to be addressed before publication.
- A thorough revision at the linguistic and editorial levels is needed.
- The statistical methods used to determine significance (p-values) are mentioned in passing but not described. It would be helpful to specify whether ANOVA, t-tests, or other methods were used. Additionally, although the introduction mentions the existence of cultivar-specific stress responses, the manuscript does not clearly state which quinoa variety was used for the experiments. This information is important, especially for assessing reproducibility and applicability.
Response to Reviewer 1 Comments:
Thank you very much for taking the time to review this manuscript. Please find the detailed responses below and the corresponding revisions highlighted changes in the re-submitted files
- Thank you for pointing this out. We agree with this comment. Therefore, the manuscript has undergone professional editing through the MDPI English Editing Service, with the certificate provided alongside this submission.
- Thank you for pointing this out. We agree with this comment. Therefore, we have added specific statistical methods in Section 4.8 (Lines 409-416) and specified the quinoa cultivar in Section 4.1 (Line 336) of the revised Materials and Methods.

Reviewer 2 Report
Comments and Suggestions for Authors
Please find attached.
Also, the introduction is deficitary, please expand with a few paragraphs on the topic of genes of abiotic stress in plants in general

Author Response
Reviewer 2 Comments:
Minor grammatical errors and typos are present (e.g., , "undersuitable" , "Pairwise variation value 2" in Figure 4b, "geNorm44,977" ). A thorough proofread is recommended. Ensure consistent formatting and clarity across all figures. For example, in Figure 2, the axes labels and legends are a bit crowded and could be optimized for better visual understanding. While the paper acknowledges discrepancies between algorithms (e.g., GeNorm and RefFinder stability rankings) , a deeper discussion on the implications of these discrepancies and how the chosen approach (RefFinder's composite ranking) addresses them would be beneficial. Also, while the paper places its findings in the context of quinoa research, expanding on the broader implications for qPCR normalization in other stress-tolerant pseudocereals or under similar abiotic stresses would strengthen the discussion.
Response to Reviewer 2 Comments:
Thank you very much for taking the time to review this manuscript. Please find the detailed responses below and the corresponding revisions highlighted changes in the re-submitted filesThank you for pointing this out. We agree with this comment. Therefore, we have proofread and revised minor grammatical errors and typos are present.
- Thank you for pointing this out. We agree with this comment. Therefore, we have revised all figure to ensure consistent formatting and clarity.
- Thank you for pointing this out. We agree with this comment. Therefore, we have re-described how RefFinder integrates multiple algorithms, with the added content located in the Discussion section (Lines 280-284). “It integrates the currently available major computational programs (geNorm, Normfinder, BestKeeper, and the comparative ΔCt method) to compare and rank the tested candidate reference genes. Based on the rankings from each program, It assigns an appropriate weight to an individual gene and calculated the geometric mean of their weights for the overall final ranking.”
- Thank you for pointing this out. We agree with this comment. Therefore, we have expanded the discussion on abiotic stress in Chenopodiaceae plants (Lines 241-247). In addition to, our future research will prioritize reference gene validation across multiple stress-tolerant pseudocereals to expand the significance of crop abiotic stress tolerance studies.

Reviewer 3 Report
Comments and Suggestions for Authors
This manuscript comprehensively evaluated reference genes for RT-qPCR normalization in Chenopodium quinoa under abiotic stress conditions, including cold, heat, and drought. Using multiple widely accepted statistical algorithms, the authors employed a gradient stress design and assessed 11 candidate reference genes (geNorm, NormFinder, BestKeeper, ΔCt method, and RefFinder). The authors also further validated the performance of the selected genes using representative stress-inducible genes (COR72, HSP44, COR413-PM, and DREB12). However, the following major and minor issues should be addressed to improve the quality of the manuscript:
1. Some candidate genes were selected based on prior transcriptomic data, but the origin of this data (conditions, developmental stage, genotype) is not described in detail. Please clarify these in the methods section.
2. The manuscript mentioned 3 biological replicates, but no information was given on technical replicates or batch effects in qPCR. Adding this information to strengthen the methodology of the study.
3. The validation using stress-responsive genes is appropriate. However, it would strengthen the manuscript to briefly explain why these specific target genes were chosen (for example, their known role in quinoa or stress response pathways).
4. Add statistical comparisons (e.g., ANOVA or t-tests) between normalization strategies to quantitatively show how reference gene choice impacts interpretation.
5. Figures:
- Please add the X-axis title for Figure 2.
- Separate Y-axis titles for figures 3a&b; 3c&dD. Space between Global, LT, HT, and DR and the figure. Add the X-axis title for Figure 3.
- Add the X-axis title for Figure 4b.
- Add Y-and X-axis titles for Figure 5a, b, c, and d.
- Figure legends: Please add detailed information, such as interpreting the abbreviation of the treatments, statistical analysis, ... for each figure legend.
6. Several candidate reference genes (e.g., SSU32, WLIM96) have unconventional names. Please provide the full protein name, gene family, and known function, if available. Including this information alongside primer details in Table 3.
Author Response
Reviewer 3 Comments:
However, the following major and minor issues should be addressed to improve the quality of the manuscript:
- Some candidate genes were selected based on prior transcriptomic data, but the origin of this data (conditions, developmental stage, genotype) is not described in detail. Please clarify these in the methods section.
2. The manuscript mentioned 3 biological replicates, but no information was given on technical replicates or batch effects in qPCR. Adding this information to strengthen the methodology of the study.
3. The validation using stress-responsive genes is appropriate. However, it would strengthen the manuscript to briefly explain why these specific target genes were chosen (for example, their known role in quinoa or stress response pathways).
4. Add statistical comparisons (e.g., ANOVA or t-tests) between normalization strategies to quantitatively show how reference gene choice impacts interpretation.
5. Figures:
- Please add the X-axis title for Figure 2.
- Separate Y-axis titles for figures 3a&b; 3c&dD. Space between Global, LT, HT, and DR and the figure. Add the X-axis title for Figure 3.
- Add the X-axis title for Figure 4b.
- Add Y-and X-axis titles for Figure 5a, b, c, and d.
- Figure legends: Please add detailed information, such as interpreting the abbreviation of the treatments, statistical analysis, ... for each figure legend.
6. Several candidate reference genes (e.g., SSU32, WLIM96) have unconventional names. Please provide the full protein name, gene family, and known function, if available. Including this information alongside primer details in Table 3.
Response to Reviewer 3 Comments:
Thank you very much for taking the time to review this manuscript. Please find the detailed responses below and the corresponding revisions highlighted changes in the re-submitted files
- Thank you for pointing this out. We agree with this comment. Candidate genes were selected based on prior transcriptomic data sourced from NCBI. We have now added the transcriptomic data identifiers in Section 4.8 'Data processing and accessibility' (Lines 409-416)
- Thank you for pointing this out. We agree with this comment. Therefore, we have added the information to the manuscript. Three technical replicates to strengthen the methodology of the study. (Line 411)
- Thank you for pointing this out. We agree with this comment. We confirm that candidate genes were selected according to their established functions in model species, given the limited characterization of these genes in quinoa. Comparative functional analyses across species are detailed in the Discussion (lines 285–310).
- Thank you for pointing this out. We agree with this comment. The impact of reference gene selection on data interpretation is addressed in the Discussion (Lines 316-324). Supplementary Table 6 provides detailed statistical validation, with significance assessed by one-way ANOVA followed by LSD tests (letters denote significant differences at p < 0.05).
- Thank you for pointing this out. We agree with this comment. Therefore, we have added key details to the X/Y-axis labels in the figures and revised all figures to ensure consistent formatting and enhanced clarity. We have added detailed information for Figure legends (Figure 2: lines 148-151; Figure 3: lines 175-178; Figure 4: lines 200-203; Figure 5: lines 232-235;).
- Thank you for pointing this out. We agree with this comment. Therefore, we have completed the revisions to Table 3 by adding full protein names and functional annotations alongside the primer details, as requested.

Round 2
Reviewer 3 Report
Comments and Suggestions for Authors
Thank you for providing the revision. The authors revised and responded to all issues positively. The manuscript was much improved, so it can be accepted for publication.